# Organizational models and patient-reported outcomes for palliative care across five tertiary hospitals in Nigeria: An environmental scan

Ann Abiola Ogbenna[1,2]*, Matthew Caputo[3], Tonia C. Onyeka[4,5], Debora O. Ohanete[1], Lyra S. Johnson[6], Nadia A. Sam-Agudu[7,8,9], Chisom Obiezu-Umeh[10], Babatunde Akodu[11,12], Denise Drane[13], Charlesnika T. Evans[14], Mukaila O. Akinwale[15], Geraldine U. Ndukwu[16], Israel K. Kolawole[17], Saheed A. Ayilara[18], Gracia K. Eke[19], Adeseye M. Akinsete[20], Adeboye Ogunseitan[21], Ashti Doobay-Persaud[3,22]

1 Department of Haematology and Blood Transfusion, College of Medicine, University of Lagos, Lagos, Nigeria, 2 Department of Haematology and Blood Transfusion, Lagos University Teaching Hospital, Lagos, Nigeria, 3 Robert J. Havey, MD Institute for Global Health, Northwestern University Feinberg School of Medicine, Chicago, Illinois, United States of America, 4 Department of Anaesthesia/Pain and Palliative Care Unit, University of Nigeria Teaching Hospital, Ituku-Ozalla, Enugu, Nigeria, 5 IVAN Research Institute, Enugu, Nigeria, 6 Program for Public Health, Northwestern University, Chicago, Illinois, United States of America, 7 International Research Center of Excellence, Institute of Human Virology Nigeria, Abuja, Nigeria, 8 Department of Pediatrics, Global Pediatrics Program and Division of Pediatric Infectious Diseases, University of Minnesota Medical School, Minneapolis, Minnesota, United States of America, 9 Department of Paediatrics and Child Health, University of Cape Coast School of Medical Sciences, Cape Coast, Ghana, 10 Department of Medical Social Sciences, Center for Dissemination and Implementation Science, Northwestern University Feinberg School of Medicine, Chicago, Illinois, United States of America, 11 Department of Family Medicine, Lagos University Teaching Hospital, Lagos, Nigeria, 12 Department of Community Health and Primary Care, College of Medicine, University of Lagos, Lagos, Nigeria, 13 Program Evaluation Core, Northwestern University, Evanston, Illinois, United States of America, 14 Center for Health Services and Outcomes Research and Department of Preventive Medicine, Institute for Public Health and Medicine, Northwestern University Feinberg School of Medicine, Chicago, Illinois, United States of America, 15 Department of Anaesthesia, College of Medicine, University of Ibadan/University College Hospital, Ibadan, Nigeria, 16 Family Medicine Department, Palliative Care Unit, University of Port Harcourt, Rivers State, Nigeria, 17 Department of Anaesthesia, University of Ilorin, Ilorin, Nigeria, 18 Department of Pain and Palliative Medicine, Federal Medical Center, Abeokuta, Nigeria, 19 University of Port Harcourt Teaching Hospital, Port Harcourt, Nigeria, 20 Department of Paediatrics, College of Medicine, University of Lagos, Lagos, Nigeria, 21 Department of Medicine, Section of Palliative Care, Division of Hospital Medicine, Northwestern University Feinberg School of Medicine, Chicago, Illinois, United States of America, 22 Departments of Medicine and Medical Education, Division of Hospital Medicine, Northwestern University Feinberg School of Medicine, Chicago, Illinois, United States of America

* aaogbenna@cmul.edu.ng

## Abstract

Palliative care (PC) is an essential, effective, and affordable component of health care. Global need is rising, with the greatest burden in low-and-middle-income countries. This is especially true in Nigeria where the need is growing rapidly, as are PC services; however, current organizational models have not yet been examined. This was a cross-sectional, descriptive study of five PC sites at tertiary hospitals in four of Nigeria's six geopolitical zones. Surveys, informed by a Centre for Palliative Care, Nigeria (CPCN) needs assessment checklist and the Consolidated Framework for Implementation Research (CFIR), were administered at each site to leadership,

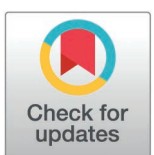

**Data availability statement:** Most data collected is contained within the tables and figures of this manuscript. Other data are in supplementary files. Please email aogbenna@unilag.edu.ng or caput044@umn.edu for any requests.

**Funding:** This work was supported by a philanthropic gift to the Robert J. Havey, MD Institute for Global Health at Northwestern University Feinberg School of Medicine (to ADP). The funder had no role in study design, data collection and analysis, decision to publish, or preparation of the manuscript. No authors received salary from the funders.

**Competing interests:** The authors have declared that no competing interests exist.

frontline workers, patients, and caregivers. Surveys varied by participant group and inquired about organizational models and personal experiences of both providers and recipients of care. Across five sites, there was a total of 282 survey respondents: five leaders, nine frontline workers, 132 patients, and 136 caregivers. The most common diagnoses of PC patients were cancer, sickle cell disease, and HIV. Most sites reported sub-optimal administrative support (80%), hospital management support (60%), and building space (60%). Leadership responses highlighted variations in PC training requirements and opportunities. Frontline workers desired additional training, sponsorship, and governmental support. Most patients and their caregivers reported satisfaction with PC, though high levels of worry and hopelessness were reported. Increased organizational support appears necessary to facilitate improvements in administrative resources, staffing, and training. Emotional and spiritual wellbeing likely require prioritization when designing palliative care delivery services in Nigeria. Further research is needed to refine current services and inform implementation efforts.

## Introduction

In 2020, global estimates indicated that more than 61 million individuals would benefit from palliative care (PC), including those in their final year of life [1]. This number is expected to rise between 2020 and 2040 [2], which is particularly concerning for low- and middle-income countries (LMICs) where 80% of the global PC burden lies [3–5]. LMICs, particularly in the regions across Africa, often face the dual challenge of increasing burden of non-communicable diseases like cancer, diabetes, cardiovascular diseases and chronic kidney disease, and limited health infrastructure to treat them [6]. The related symptomatology due to this increase in non-communicable diseases underscores a critical need for comprehensive strategies to integrate PC into healthcare systems in these regions [7].

PC alleviates suffering for patients with life-limiting illnesses and their caregivers by reducing physical, psychosocial, and spiritual symptoms at any stage of their condition [8]. The WHO asserts that PC is a human right and calls for its inclusion in universal health coverage, recognizing the need for global efforts to improve PC services and build workforce capacity in resource-constrained settings [9].

In recent years, there has been an expansion of PC services across Africa with 42% of countries having a focal person in their ministry designated for PC [10], 54% having a national PC association, and over 1000 services available across the continent [10,11]. However, growth has been uneven with regional concentrations in East and Southern Africa, and services [12] remain accessible to less than 5% [12] of the nearly 9.7 million people that need it across the African continent [3,13]. This has left significant care gaps in West African countries like Nigeria where the need for PC is critical. It is estimated [14]. that 29% of the nearly 3 million annual deaths [15] across Nigeria are due to non-communicable diseases such as cardiovascular diseases and

cancers, and over 50,000 are due to HIV [16]. For cancer patients specifically, most present in advanced stages, report high levels of pain [17] and have little hope for a cure [18].

Unified efforts to promote provision of PC in Nigeria include the establishment of the Hospice and Palliative Care Association of Nigeria (HPCAN) in 2007, which has facilitated the establishment of eight PC centres at tertiary health institutions and over 20 emerging PC sites as of 2016. In 2021, the Federal Ministry of Health published a National Policy and Strategic Plan for Hospice and Palliative Care [19] with the goal of institutionalizing PC across all levels of the health system. While the increasing number of services is recognized, there are few comprehensive studies that describe operational characteristics of these PC sites, limiting our understanding of the current state of PC in Nigeria and impeding efforts to improve, enhance and expand service [20]. Specifically, transformational change in PC access and delivery requires greater organizational knowledge relating to infrastructure, workforce, service delivery, and satisfaction of care [12,21]. Through an environmental scan, our study aims to describe the current models of PC in select tertiary hospitals across Nigeria and examine patient-centered outcomes under these models of care.

## Methods

### Ethics statement

Ethical approval for this study was obtained from the National Human Research Ethics Committee of Nigeria (NHREC) [Approval No: NHREC/01/01/2007] and from the institutional Health Research Ethics Committees (HREC) of participating institutions. Verbal informed consent was obtained from all participants after a clear explanation of the study's purpose, procedures, potential risks, and benefits. The use of verbal consent was approved by the ethics committees due to the minimal risk nature of the study and the cultural acceptability of verbal agreements in the study context. No name was recorded. All information gathered was confidential and strictly used for academic purposes only. Access to data obtained was restricted to the principal investigator and delegated persons. Participation in the study was entirely voluntary and participants were free to withdraw from the study at any time with no implication to respondents' care or care of their loved ones. No financial responsibility was borne by the participants.

### Study design

This was a cross-sectional, descriptive study of select PC sites at tertiary hospitals in four (North-Central, South-West, South-East, and South-South) of Nigeria's six geopolitical zones. The recruited sites belonged to a consortium formed to design and implement a series of virtual PC training sessions for providers in 2021 [22,23]. These five sites were located in distinct Nigerian cities (Enugu, Ilorin, Lagos, Port Harcourt, and Abeokuta). REDCap surveys were administered in PC units at each site from February to April 2023 to assess existing organizational models and patient-centered outcomes.

### Study population

Unique surveys were administered to four groups at each site: leadership, frontline workers, patients, and caregivers. Leaders were defined as administrators or physicians heading the PC units at each site. Frontline workers were defined as physicians, nurses, and social workers working in the PC units. Patients were defined as individuals actively receiving PC services, and caregivers were defined as friends or family who supported the patients. All participants were 18 years or older. Recruitment of participants started on the 15th of February 2023 and ended on the 13th of April 2023

### Surveys

Leadership and frontline worker surveys included checkbox, multiple choice, and free-response questions. Leadership surveys aimed to capture information about PC infrastructure, operations, staffing, training, and patient characteristics. Frontline worker surveys asked about prior PC training, areas where more training was desired, and two open-ended

questions: 1) "What do you expect from a good palliative care team?", and 2) "Is there anything else you would like to communicate about this subject of palliative care?" The surveys were informed by a needs assessment for the establishment of PC developed by the Centre for Palliative Care in Nigeria (S1 Text) and the Consolidated Framework for Implementation Research (CFIR) [24]. CFIR is a widely cited implementation research framework that helps to identify barriers and facilitators to effective implementation through five domains (Innovation, Outer Setting, Inner Setting, Individuals, and Implementation Process) and 39 underlying constructs [25]. In the present study, CFIR was employed to 1) construct and prioritize survey questions, and 2) organize the questionnaire, in order to identify determinants of PC implementation in the Nigerian setting. Shortly following the development of our surveys in 2022, CFIR 2.0 was published with revised domains and constructs [26]. We considered the updated framework during analysis.

Patient and caregiver surveys aimed to reflect the quality of PC services through assessing patient-centered outcomes in the 3 days prior to survey collection. The patient surveys (7 items) and caregiver surveys (3 items) contained 5-point Likert scale questions from the African Palliative Outcome Scale, a validated measure developed by the African Palliative Care Association [27,28]. Clinical history of symptomatology and impact of PC on its outcome was assessed through the African Palliative Outcome scale. All participants were above the age of 18 years. A proportionate sampling was used to determine target sample sizes at each site based on the approximate number of active patients at each site. Final samples per site ranged from 13 to 53 patients and 13–53 caregivers. A convenience sampling technique was used, and none of the participants declined to participate.

A summary of the surveys can be found in -S2 Text.

### Data collection, management, and analysis

All surveys were designed and administered via REDCap from February to April 2023. A research assistant was recruited from the PC department at each site and trained to administer surveys online through a QR code or URL. Translation and/or verbal administration of the surveys were provided in Yoruba and Igbo (the dominant languages in the site locations) for patients and caregivers who did not speak English or were unable to read. Instruments were not pre-translated ahead of time; however, research assistants were selected from PC care staff who were accustomed to translating or discussing issues of PC with patients in their local languages.

Data were analyzed using descriptive statistics, including counts (proportions) and medians (with min-max ranges). Staff-to-patient ratios for each staff cadre were calculated at each site by dividing the number of staff members by the reported patients per week. Patient and caregiver responses to Likert questions were weighted so that each site, regardless of recruitment size, contributed to the aggregate results equally. Mean (SD) ratings were calculated by site, and ANOVA tests for equal means were performed for each question.

Two coders conducted a rapid qualitative review of frontline workers' open-ended responses.. Representative quotes were selected to reflect summarized consensus responses that had been organized according to CFIR 2.0 domains and specific constructs. Statistical analyses were performed in R version 4.3.1 [29], with visualizations generated with the *sjPlot* package and Microsoft Excel.

## Results

There was a total of 282 survey respondents: five leaders, nine frontline workers, 132 patients, and 136 caregivers. A breakdown of participants by site is shown in Table 1.

### Leadership surveys

Of the surveyed leaders, five (100%) reported providing outpatient services, three (60%) reported providing inpatient services that were consultative or primarily for PC admission, and three (60%) reported providing inpatient hospice care. Characteristics of site resources, capacity, and operations can be found in Table 2. All leaders reported existing data

**Table 1. Survey respondents by site.**

|  | Leaders | Frontline Workers | Patients | Caregivers | Total |
|---|---|---|---|---|---|
| **Site 1** | 1 | 3 | 13 | 15 | 32 |
| **Site 2** | 1 | 1 | 13 | 13 | 28 |
| **Site 3** | 1 | 3 | 13 | 14 | 31 |
| **Site 4** | 1 | 1 | 40 | 41 | 83 |
| **Site 5** | 1 | 1 | 53 | 53 | 108 |
| **Total** | 5 | 9 | 132 | 136 | 282 |

**Table 2. Site settings, operations, resources, and support.**

| Characteristic | Response or Count | | | | | Median (Min-Max Range) or Count (%) |
|---|---|---|---|---|---|---|
|  | Site 1 | Site 2 | Site 3 | Site 4 | Site 5 | Total |
| Year site established |  |  |  |  |  | 2008 (2001–2012) |
| Unit co-located with the main hospital center | Yes | Yes | Yes | Yes | Yes | 5 (100%) |
| Stand-alone unit | Yes | Yes | Yes | Yes | Yes | 5 (100%) |
| Designated PC wards | No | No | No | Yes | No | 1 (20%) |
| Have a room designated for: |  |  |  |  |  |  |
| PC pharmacy | Yes | No | No | No | No | 1 (20%) |
| PC medical records | Yes | No | No | Yes | Yes | 3 (60%) |
| # PC consulting rooms | 1 | 2 | 2 | 1 | 4 | 2 (1–4) |
| # PC treatment rooms | 1 | 1 | 1 | 1 | 1 | 1 (1–1) |
| Capacity of PC waiting room | 4 | 50 | 5 | 20 | 15 | 20 (4–50) |
| Housing system for patients' relatives | No | No | No | No | No | 0 (0%) |
| Have the following tools/resources: |  |  |  |  |  |  |
| Enrollment cards | Yes | Yes | Yes | Yes | Yes | 5 (100%) |
| Tablets/computers | No | No | No | Yes | No | 1 (20%) |
| PC appointment system | Yes | No | Yes | Yes | No | 3 (60%) |
| Palliative care services offered |  |  |  |  |  |  |
| Inpatient services | Yes | Yes | Yes | Yes | Yes | 5 (100%) |
| Outpatient services | Yes | Yes | Yes | Yes | Yes | 5 (100%) |
| Home-based services | Yes | Yes | Yes | Yes | Yes | 5 (100%) |
| For profit | No | No | No | Yes | No | 1 (20%) |
| Sponsored/Major donor | No | No | No | No | No | 0 (0%) |
| Report adequate PC space or building | No | Yes | No | Yes | No | 2 (40%) |
| Feel well supported by hospital | Disagree | Agree | Neutral | Neutral | Agree | – |
| Feel they receive adequate administrative support | Neutral | Agree | Neutral | Neutral | Neutral | – |

A stand-alone unit implies the PC service is provided as an independent functional unit within the hospital. Designated PC wards imply that the wards are designated only for patients with PC needs. Inpatient PC services are consultative services, in which the patient is admitted by his or her primary clinician, who now refers to the PC unit for review or to co-manage, while a PC admission is when the patient is admitted primarily by the PC physician for primarily PC.

collection procedures for patient outcomes and research, while four (80%) reported these for administrative reporting and quality improvement.

All PC sites were established between the years 2001 and 2012. The distribution of staffing at each site varied and is visualized in "Fig 1".

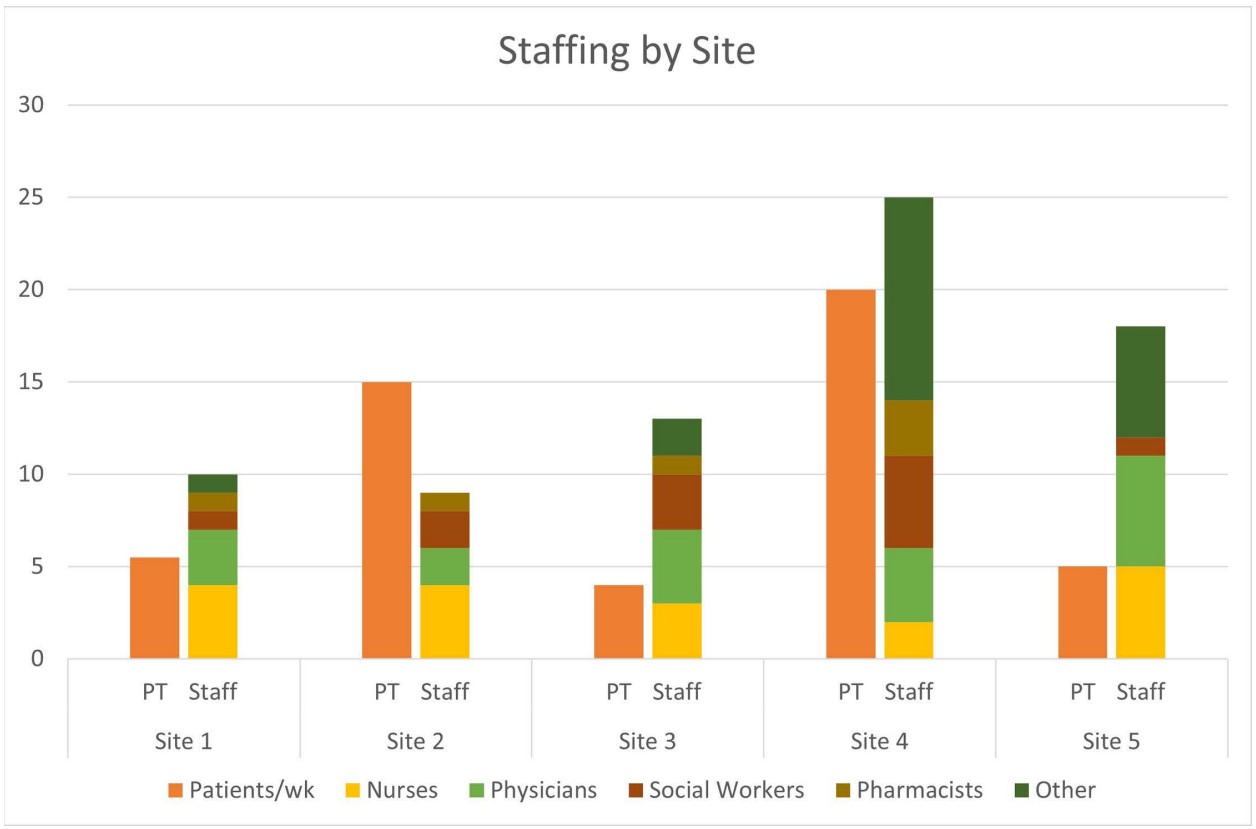

**Fig 1. Staffing composition and patient volume across five clinical sites.** The chart compares the average number of patients per week (orange bars labeled "PT") and the total staff composition (stacked bars labeled "Staff") across five clinical sites. Staff are categorized by professional roles: nurses (yellow), physicians (light green), social workers (brown), pharmacists (mustard yellow), and other personnel (dark green).

Other identified staff included physiotherapists, clergy, and administrative staff. All sites reported treating both adult and pediatric patients, with the most common illnesses being cancer, sickle cell disease, and HIV. Leaders reported receiving the most referrals from the following departments: Surgery; Radio-Oncology; Oncology; Paediatrics; Gynaecology; Ear, Nose and Throat; and Internal Medicine (Table 3).

All sites reported having PC seminars for their staff, with three (60%) having them weekly and two (40%) having the semi-annually or less often. One site (20%) reported that all staff members had some type of formal training in PC while the 4 other sites (80%) had some staff with formal training and others with informal training. Of the five leaders, three (60%) reported plans for certificate training and four (80%) reported plans for conference attendance.

**Frontline worker surveys**

Of the nine frontline workers surveyed, five (56%) reported having PC training ever, including two (22%) that were trained in the past two years. When asked in which areas of PC they felt they needed more training (check all that apply), seven (78%) indicated advance care directives, five (56%) indicated communicating bad news, five (56%) indicated pain management, five (56%) indicated symptom management and three (33%) indicated assessing the goal of care.

Many opinions from providers emerged surrounding the qualities of an effective PC team and the state of PC in Nigeria (Table 4). Frontline workers highlighted the collaborative and multidisciplinary nature of the PC team and emphasized the importance of honesty, empathy, and communication skills. One respondent described PC as "still in infancy" in Nigeria,

**Table 3. Site staffing and patient census.**

| Characteristic | Response or Count | | | | | Median (Min-Max Range) or Count (%) |
|---|---|---|---|---|---|---|
| | Site 1 | Site 2 | Site 3 | Site 4 | Site 5 | Total |
| No. of outpatient clinic days per week | 1 | 5 | 1 | 2 | 7 | 2 (1–7) |
| Patients per week | 4-7 | 15 | 4 | 20 | 4-6 | 5.5 (4–20) |
| Referrals per week | 4-6 | 4-6 | 2-3 | 7+ | 4-6 | 5 (2–7+) |
| Total PC staff per site: | 10 | 9 | 13 | 25 | 18 | 13 (9–25) |
| Nurses | 4 | 4 | 3 | 2 | 5 | 4 (2–6) |
| Physicians | 3 | 2 | 4 | 4 | 6 | 4 (2–6) |
| Social Workers | 1 | 2 | 3 | 5 | 1 | 2 (1–5) |
| Pharmacists | 1 | 1 | 1 | 3 | 0 | 1 (0–3) |
| Other | 1 | 0 | 2 | 11 | 6 | 2 (0–11) |
| Other staff specified | Physio-therapist | Physio-therapist | Clergy, Health Assistant | Did not specify | Medical Records, Administrative Staff | – |
| Treat Both Adults and Children | Yes | Yes | Yes | Yes | Yes | 5 (100%) |
| Most common adult patients | Cancer, HIV | Cancer (breast, prostate, liver) | Geriatric Cancer | Cancer, HIV, Sickle Cell | Cancer (breast, prostate, cervical) | – |
| Most common pediatric patients | Cancer, Sickle Cell, HIV | Burkitts lymphoma, retino-blastoma, sickle cell | Cancer, Sickle Cell, HIV | Cancer, Sickle Cell, end stage organ dx | Sickle cell, Nephro-blastoma, lymphoma | – |

**Table 4. Frontline worker representative quotes.**

| Frontline Worker Quote | Summary Consensus Response | CFIR Domain: Construct(s) |
|---|---|---|
| *"A good Palliative Care team should be multidisciplinary comprising of the relevant health care workers and religious leaders. The approach should be holistic, taking care of the total man"* | PC teams should be multidisciplinary, including religious leaders, to provide holistic care | **Inner Setting:** Work Infrastructure, Culture, Recipient-Centeredness<br>**Implementation Process:** Teaming |
| *"They [PC team] are to advocate for them, Have good communication skill and work together to achieve the set goals for the patients. They ought to be honest and trusted."* | Members of PC teams should work together to achieve the goals set for the patients. They should be honest, trusted, and have effective communication skills. The PC team should be advocates for the patients. | **Inner Setting:** Recipient-Centeredness, Communication |
| They [PC team] should be very sympathetic and emphatic towards those in pain. Showing love, support and care to them. Always available to relieve someone in pain and family. | The PC team should be sympathetic and empathetic. They should show love, support, and care to the patient. Whenever needed, the PC team should provide relief for both patients and their family members. | **Inner Setting:** Recipient-Centeredness |
| *"Palliative Care is still in infancy in Nigeria. Creation of awareness, education and training of health care workers who are interested in the field should be encouraged."* | PC is relatively new to Nigeria. There is a need to spread awareness of PC among healthcare workers in Nigeria. Education and training should be encouraged for healthcare workers interested in the field. | **Outer Setting:** Local Attitudes<br>**Inner Setting:** Access to Knowledge and Information |
| *"Advocacy, Policy making and Implementation of palliative care should start from government. Education and Training of personnel to advanced level is also germane."* | Nigerian PC providers believe their government should make efforts to advocate for PC, include it in policy, and promote implementation. There is a need for advanced education and training for PC. | **Outer Setting:** Policies and Laws<br>**Inner Setting:** Tension for Change, Relative Priority<br>**Individuals Involved:** capacity building |

and multiple respondents expressed a need to spread awareness and advocate for this type of care. Frontline workers also indicated a need for additional training, sponsorship, and governmental support.

**Patient and caregiver outcomes**

The following percentages reflect African Palliative Outcome Scale responses of 4 or 5 on the 5-point Likert scale with equal weighting by site "Fig 2".

On average, sites found that 21% of their patients reported high levels of pain (POS 1) "Fig 2", 9% were highly affected by other symptoms (POS 2) "Fig 2", and 31% were highly worried about their illness (POS 3). Nearly 68% of patients reported being able to share their feelings freely (POS 4) "Fig 2", and 75% had enough advice to plan for the future (POS 7) "Fig 2", although only 38% felt life was worthwhile (POS 5) "Fig 2", and 44% felt at peace (POS 6) "Fig 2",. On average, 40% of the caregivers reported high levels of worry about the patient's illness (POS 10) "Fig 3".

About 75% of caregivers felt that they had been given sufficient information (POS 8) and felt confident caring for the patient (POS 9) "Fig 3". ANOVA tests revealed that at least one site's average rating differed significantly from the others for POS 2 ($p = .001$), POS 4 ($p = .003$), POS 7 ($p < .001$), and POS 9 ($p < .001$) (S3 Text).

## Discussion

PC in Nigeria has been slowly evolving, and it is important to characterize the state and scope of services to understand what is needed for scale up and improved impact. Key findings from this study include multidisciplinary approaches to PC management despite challenges with administrative support, funding, staffing, and training. Providers emphasized the importance of providing holistic care with empathy and indicated the need to advocate for PC scale-up in the Nigerian health system. Patient-reported outcomes highlight the need to improve services for pain management as well as emotional and spiritual wellbeing.

Nurses, physicians, and social workers staffed each unit and reflected the hospital consultative team model, which is the predominant model for PC delivery in the United States [30]. In Nigeria, health staffing shortages have been worsened by the intensification of the 'Japa syndrome' that has seen nurses, physicians and other healthcare workers migrate to greener pastures for enhanced compensation, upgraded working conditions, and progression in career, this may be a potential barrier to PC delivery [30,31].

All but one site (which had the highest average pain rating) reported having a pharmacist on staff for their PC patients, which was a positive finding as accessibility of opioids and an inadequate prescribing workforce are major barriers to PC provision in Africa [32,33]. This may explain why most patients in our study reported neutral or favorable responses when asked if they had recently been in pain or had experienced other affective symptoms, which differs from a previous finding that pain was the most poorly rated item on the African Palliative Outcome Scale and most frequent complaint across many studies [32]. This may be attributed to the presence of pharmacists and likely consequential availability of opioids [34] at all but one of the included sites. However, since our surveys did not specifically ask about access to analgesics, further inquiry is needed. Additionally, while pain ratings were lower than expected, the finding that over 20% of patients on average reported high levels of pain (4 or 5 out of 5) in their last 3 days of care warrants further investigation into root causes for poor pain control, including access to pain medicine, use of opioids, patient communication of pain, prescribing practices, and utilization of physical therapy.

Court *et al.*'s systematic review of PC integration in Africa emphasized that adequate training within sufficiently sized staff is a significant facilitator of PC integration [35]. In our limited sample size of nine frontline workers, about half had some type of PC training, while only 22% had training in the last two years. Barriers to sufficient staffing of trained personnel are exacerbated by general limitations in human resources in African contexts [36] and limited knowledge of PC in the existing workforce [32], emphasizing the need to train and retain existing providers and equip them to disseminate PC knowledge and skills in their local contexts. In our study, responses to open-ended questions by frontline workers stressed

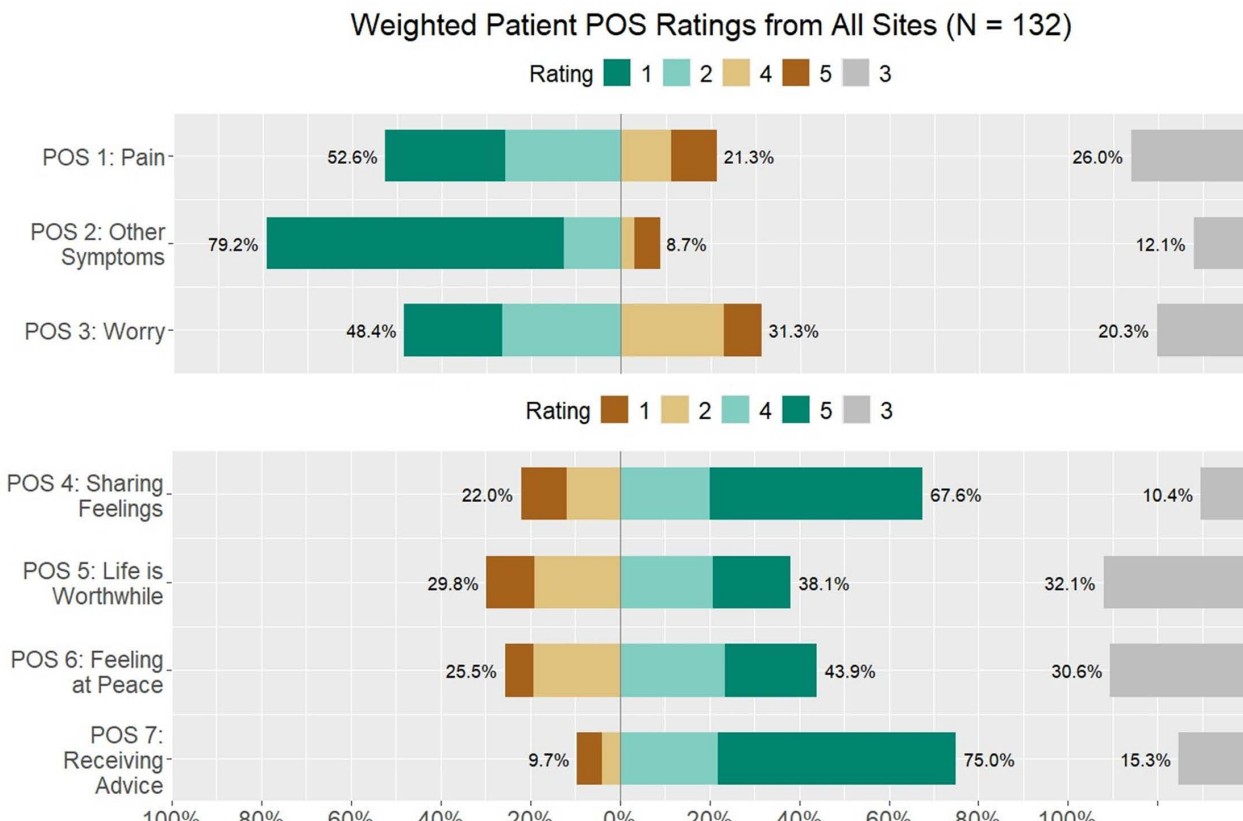

### Weighted Patient POS Ratings from All Sites (N = 132)

| Item | Description | Rating: 1 | Rating: 5 |
|------|-------------|-----------|-----------|
| POS 1 | Please rate your pain during the last 3 days | No Pain | Worst / Overwhelming |
| POS 2 | Have any other symptoms (e.g. nausea, coughing or constipation) been affecting how you feel in the last 3 days? | Not at all | Overwhelmingly |
| POS 3 | Have you been feeling worried about your illness in the past 3 days? | Not at all | Overwhelming worry |
| POS 4 | Over the past 3 days, have you been able to share how you are feeling with your family or friends? | Not at all | Yes, I've talked freely |
| POS 5 | Over the past 3 days have you felt that life was worthwhile? | Not at all | Yes, all the time |
| POS 6 | Over the past 3 days, have you felt at peace? | Not at all | Yes, all the time |
| POS 7 | Have you had enough help and advice for your family to plan for the future? | Not at all | As much as wanted |

**Fig 2. Weighted patient-reported outcomes from all clinical sites using the Palliative Outcome Scale (POS).** Responses from 132 patients were aggregated across seven POS items assessing physical, psychological, and emotional well-being over the previous three days. Bars show the distribution of responses rated from 1 to 5. Items cover pain (POS 1), other symptoms (POS 2), worry (POS 3), ability to share feelings (POS 4), sense of life being worthwhile (POS 5), feeling at peace (POS 6), and receiving advice (POS 7). The table below the chart explains each POS item and anchors for Ratings 1 and 5.

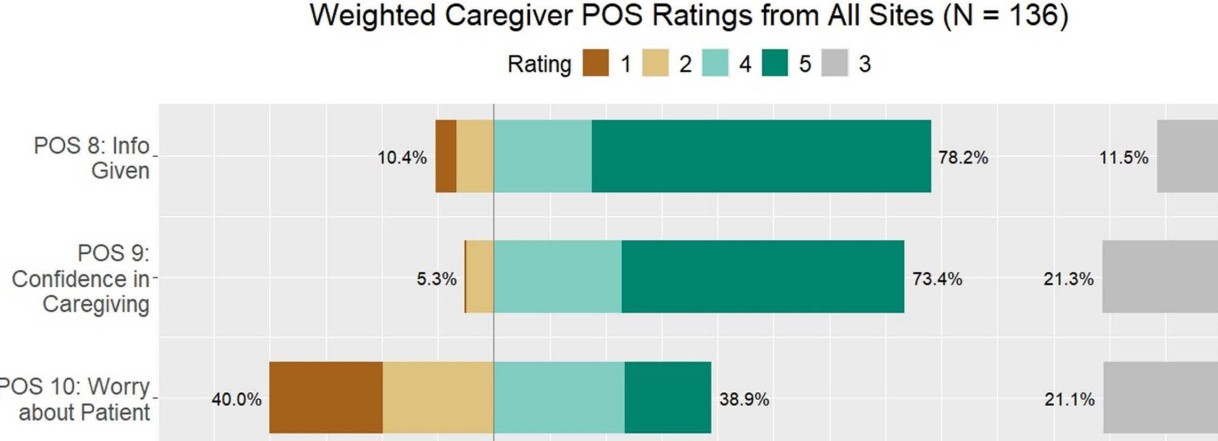

**Fig 3. Weighted caregiver ratings of Patient-Oriented Scale (POS) items across all sites (N = 136).** The figure displays the distribution of caregiver responses (ratings 1–5) for three key POS items: information provided (POS 8), confidence in caregiving (POS 9), and worry about the patient (POS 10). Percentages indicate the proportion of caregivers selecting each rating category, with higher ratings (e.g., 5) reflecting more positive outcomes. Data are presented as weighted aggregates from all participating sites.

the importance of training and expressed the desire to receive more of it. Feasibility for such training in Africa has been demonstrated through successful initiatives by the African Palliative Care Association [37] and, in Nigeria, a collaboration between Lagos University Teaching Hospital and Northwestern University [22,23].

Most sites in our study reported inadequate administrative support, hospital management support, and building space -- a concerning trend as support from health systems is a key determinant in the effective integration of PC services in Africa [20]. This may stem from minimal awareness or misconceptions about PC among hospital managements as multiple participants expressed its state of infancy in the Nigerian health system. Similar sentiments were reflected in a qualitative study of newly qualified physicians in Nigeria, where some were entirely unaware of PC units existing at their institutions [38]. These shortcomings must be addressed as healthcare worker knowledge, acceptability, motivation and commitment are key sub-themes in health systems strengthening to support PC [35]. In a systematic review by Court and Olivier [35], they found that advocacy from PC associations, champions of PC teams, and PC patients [39], and faith-based community structures [40] were particularly effective in improving acceptability and prioritizing PC within health systems in African countries. Thus, we believe that advocacy, aimed at both local healthcare professionals and leaders at the hospital and governmental level, will be a critical step in improving support for existing and future PC services.

Open-ended survey responses from frontline workers provided insight into their perceptions of successful PC. Participants described the interdisciplinary and holistic aspects of PC, alluding to the interplay of medical, social work, and

spiritual support that comprise recommended PC models [9]. Previous studies indicate that spiritual care may be particularly relevant to African PC patients. A review of PC studies across Africa highlighted the implications of a commonly "intertwined perception of spirituality and religion", reporting that patients and caregivers struggled to find meaning in the illness, and how many patients used faith and prayer to cope with worry and anxiety [41]. While chaplains are commonly involved in American PC provision [42], only one site in our study (site 3), which had the highest mean score for POS 5 (feeling life was worthwhile) and POS 6 (feeling at peace), reported having a cleric on their team, revealing a potential gap in current care models. Several participants also underscored the importance of compassion, honesty, and strong communication, demonstrating that these commonly identified facilitators of PC provision [43–45] are relevant to the Nigerian context.

Patients reported mostly favorable experiences about sharing feelings, but mostly unfavorable or neutral feelings of worry, peace, and life being worthwhile. The depth of this finding is heightened by a study of PC patients in South Africa and Uganda by Selman *et al.*, which found that feeling at peace and sensing meaning in life were more important to PC patients than physical comfort, and spiritual well-being was the outcome most correlated with overall quality of life [46]. Caregivers in our sample reported high levels of worry, which is concerning since family member caregivers of PC patients in Africa have demonstrated a high risk for depressive symptoms [47]. As PC provision [48] and decision-making [49,50]. become increasingly dependent on family members, more specific family-centered outcomes should be evaluated to identify actions or interventions that may improve the well-being of Nigerian PC caregivers. Of note, most patients and caregivers reported that they received sufficient information and advice to plan for the future, which is a commonly unmet need in PC [51].

## Limitations

There were a few limitations to the study. As this environmental scan was only administered to a geographical subset of tertiary PC sites in Nigeria, generalizability to other institutions in Nigeria is limited. As only one to three frontline workers were surveyed per site, results regarding worker knowledge and training may be biased. Also, having a mix of dedicated staff and other staff with various responsibilities in the hospital aside the provision of palliative care services will skew the staff capacity at each site. Additionally, since all participants were over the age of 18, we cannot generalize patient and caregiver outcomes to the pediatric population. While the African Palliative Outcome Scale is a validated tool, there are no validated translations into local Nigerian languages and this could be a potential source of bias. Due to the limited number of sites included in this study, we were unable to analyze associations between patient-centered outcomes and site characteristics.

## Conclusion

Our findings provide insights into the current models of PC delivery across Nigeria and the experiences of both providers and recipients of this care. Increased organizational support appears necessary to facilitate improvements in administrative resources, staffing, and training. Worry and hopelessness were prevalent among this sample of patients and caregivers, highlighting the need to prioritize emotional and spiritual wellbeing in Nigerian PC services. As the Nigerian population requiring palliation grows rapidly, further research is needed to refine current services and inform implementation efforts.

## Supporting information

**S1 Text. Checklist to assess site specific infrastructure and personnel.**
(DOCX)

**S2 Text. Environmental scan survey summaries.**
(DOCX)

**S3 Text. Palliative Outcome Scale (POS) per site.**
(DOCX)

**S4A Text. Admin/lead questionnaire.**
(PDF)

**S4B Text. Patient questionnaire.**
(PDF)

**S5 Text. Inclusivity in global research questionnaire.**
(DOCX)

**S1 Data. Palliative Outcome Scale (POS) for patients.**
(XLSX)

**S2 Data. Palliative Outcome Scale (POS) for caregivers.**
(XLSX)

## Acknowledgments

We owe tremendous thanks to the Nigerian palliative care providers, patients, and caregivers that participated in this study.

## Author contributions

**Conceptualization:** Ann Abiola Ogbenna, Babatunde Akodu, Ashti Doobay-Persaud.

**Data curation:** Matthew Caputo.

**Formal analysis:** Matthew Caputo.

**Funding acquisition:** Ann Abiola Ogbenna, Ashti Doobay-Persaud.

**Investigation:** Ann Abiola Ogbenna, Tonia C Onyeka, Ashti Doobay-Persaud.

**Methodology:** Ann Abiola Ogbenna, Nadia A. Sam-Agudu, Chisom Obiezu-Umeh, Denise Drane, Charlesnika T. Evans, Adeboye Ogunseitan, Ashti Doobay-Persaud.

**Project administration:** Ann Abiola Ogbenna, Debora O. Ohanete, Lyra S. Johnson, Mukaila O. Akinwale, Geraldine U. Ndukwu, Israel K. Kolawole, Saheed A. Ayilara, Gracia K. Eke, Adeseye M. Akinsete, Ashti Doobay-Persaud.

**Resources:** Ann Abiola Ogbenna, Mukaila O. Akinwale, Geraldine U. Ndukwu, Israel K. Kolawole, Saheed A. Ayilara, Gracia K. Eke, Adeseye M. Akinsete.

**Supervision:** Ann Abiola Ogbenna, Mukaila O. Akinwale, Geraldine U. Ndukwu, Israel K. Kolawole, Saheed A. Ayilara, Gracia K. Eke, Adeseye M. Akinsete, Ashti Doobay-Persaud.

**Visualization:** Matthew Caputo.

**Writing – original draft:** Ann Abiola Ogbenna, Matthew Caputo, Tonia C Onyeka, Lyra S. Johnson, Ashti Doobay-Persaud.

**Writing – review & editing:** Ann Abiola Ogbenna, Matthew Caputo, Tonia C Onyeka, Debora O. Ohanete, Lyra S. Johnson, Nadia A. Sam-Agudu, Chisom Obiezu-Umeh, Charlesnika T. Evans, Ashti Doobay-Persaud.

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
