## [Decision Letter · Decision Letter 0]

29 Dec 2024

PGPH-D-24-02428

Organizational models and patient-reported outcomes for palliative care across five tertiary hospitals in Nigeria: an environmental scan

Dear Dr. Ogbenna,

Thank you for submitting your manuscript to PLOS Global Public Health. After careful consideration, we feel that it has merit but does not fully meet PLOS Global Public Health’s publication criteria as it currently stands. Therefore, we invite you to submit a revised version of the manuscript that addresses the points raised during the review process.

We look forward to receiving your revised manuscript.

Kind regards,

Sakib Burza, MBChB, MRCGP, MSc, PhD

Academic Editor

Journal Requirements:

Additional Editor Comments (if provided):

Reviewers' comments:

Reviewer's Responses to Questions

**Comments to the Author**

1. Does this manuscript meet PLOS Global Public Health’s publication criteria?

Reviewer #1: Partly

Reviewer #2: Yes

2. Has the statistical analysis been performed appropriately and rigorously?

Reviewer #1: No

Reviewer #2: I don't know

3. Have the authors made all data underlying the findings in their manuscript fully available (please refer to the Data Availability Statement at the start of the manuscript PDF file)?

Reviewer #1: No

Reviewer #2: No

4. Is the manuscript presented in an intelligible fashion and written in standard English?

Reviewer #1: Yes

Reviewer #2: Yes

Reviewer #1: I would like to thank the authors for the opportunity to review this interesting study. This work addresses an important gap in the literature and supports needed development of palliative care in the West African context.

My main overall piece of feedback is that the data analysis, particularly of the patient/caregiver surveys could be more robust. There is a mention early in the paper regarding a confidence interval used to select the sample size, but then only descriptive statistics are reported. Since the authors aim to compare models of care, it would be more interesting to compare outcomes across sites even if findings are not statistically significant. Please find my other comments below:

Line 94: Typographical error: I think the authors meant "is expected to rise between 2020 and 2040"

Line 96: What is the other side of this dual challenge?

Line 135-136: The comment that all participated seems unnecessary unless other sites were invited who did not participate.

139 and 169: Please explain the sampling approach in more detail. How were participants selected? Was this a convenience sample or was random sampling used? Did any participants decline to participate?

162: This is a helpful explanation given the updates to CFIR

167: Is there a reason no clinical or demographic information were collected? This may have helped to contextualize the findings

168: Since you didn’t include any statistical inferences and only descriptive statistics in the analysis, the mention of margin of error and confidence intervals does not seem relevant here.

177: Could you say more about how the accuracy of the translations was ensured? Were the instruments translated ahead of time or translated in the moment by the research assistant? Was this person a trained translator?

183: Is there a reason you aggregated findings rather than comparing them between sites? Even if the sample size was small, this may have been useful information.

184-185: Could you say more about the methodology of this qualitative rapid review? Were data coded? How were quotations selected?

196-197: Could you define the difference between PC admission and inpatient hospice care?

Table 2: Could you clarify the difference between a standalone unit and a designated PC ward?

204-206: These ratios are somewhat hard to interpret. If I understood correctly, these same teams are involved in providing inpatient care? If so, the ratios are interesting but they don't take into account workload involved in inpatient care.

Table 3: Are all of these dedicated staff? Do they only work in the PC department/service or do they have other responsibilities?

219: I can see why including one leader per site was enough because you were focused on characteristics of the program and a single person could answer this. But a larger sample size for staff would have strengthened the findings. Perhaps you can offer an explanation for the small sample of staff (several sites only had 1 respondent).

220-221: Interesting that this contradicts what leaders shared. Did they define training differently maybe?

Table 4: Did the participant mean empathic or empathetic?

237-244: It is difficult to interpret these proportions without a comparator. It may be more useful to compare these outcomes between sites for example. Did sites with different models of care have different outcomes?

Additionally, when you say that 21% agreed they were in pain, was this a yes or no question? If not, it would be good to clarify if this includes patients who agreed as well as those who strongly agreed? And so forth for the other items.

Does the question about pain assess if the participant is currently in pain or ever in pain? Does this indicate effective pain control or a population in which pain is not the primary problem?

Overall the analysis of the patient/caregivers surveys could be more robust.

251: It actually seems that pain control was quite good, although the finding is hard to interpret.

259: As I mentioned before, it's hard to interpret this. If one nurse is seeing 10 outpatients in a week, that still seems like a good ratio, but it really depends on that nurse's other duties.

264: It would have been interesting to ask front-line staff and leaders specfiically about opioid access and use.

270: Could you address the discrepancy between how leaders and front-line staff described training coverage?

302-303: This is a helpful point about the need to incorporate spiritual support.

307-308: The phrasing here is a bit confusing. Positive in the sense that they agreed they had pain or positive in the sense that it was good that they had good pain control. Consider using "neutral or disagree" that they had been in pain.

312: This is a helpful notation of a limitation of the study. Consider mentioning that the question is ambiguous about whether this is assessing current pain, or "having been in pain" at some point.

321: In what way is decision-making increasingly dependent on family members? Are you referring to cultural trends or as the patient's condition worsens?

335: Is this due to the number of sites or the number of participants per site? It seems as though you could compare for example, the patient centered outcomes between sites that had a standalone unit, different staffing ratios, or perhaps adequate space or not?

337: It seems as though your findings are more descriptive of the challenges different teams face more so than their model of care. I would have liked to hear more about how the programs were organized (standalone, outpatient, consulting) and how that relates to patient-centered outcomes.

355: It would be helpful to mention ethics approval and consent procedures in more detail in the body of the manuscript

Reviewer #2: The manuscript aims to answer 2 questions about palliative care delivery in Nigeria, namely 1. models of palliative care delivery and 2. patient and caregiver centred outcomes for those receiving palliative care. This manuscript adds to the literature and gives insight into palliative care development in Nigeria.

The authors main conclusions are justified by the data presented in the manuscript.

A weakness in this study is that only 5 out of 8 PC centres within tertiary health institutions were surveyed. The manuscript describes PC being provided in more than 20 sites in Nigeria, and it would have been interesting to learn more about organisational models also outside of tertiary centres.

Survey participants:

The author describes the survey of frontline staff, yet the survey is not provided in the appendices. The number of frontline staff surveyed was very low, and given one of the key reported findings was a lack of training in palliative care (only 56% of staff reported to have ever received palliative care training), the author should explain if efforts were made to survey more frontline staff and if not, why not. Also note discrepancy between leadership and frontline workers reporting of training.

Data:

The author provides data of organisational models in table 2, and the leadership survey (of organisational models) is provided in appendix 1. The data in table 2 misses some key information from the survey e.g. availability of home-based palliative care. In addition, data is reported in table 2 that does not appear to be asked about in the survey included in the manuscript, e.g., housing system for patient’s relatives.

Staff ratios are reported 1:1 to 1:10 staff to patients per week, yet the location of care does not seem to be mentioned here (10 out-patients per staff member per week versus 10 in-patients?).

Interpretation of patient and caregiver POS:

The author does not include in the manuscript the timing of patient and caregiver POS, e.g., on first assessment at time of referral, or 6 weeks after receiving palliative care etc. This is relevant to interpret the impact of palliative care on the POS. Yet the author appears to reach 2 contradictory interpretations from the POS in the discussion section. See Line 251/252 – highlight need to improve services for pain management and line 307/308 – suggests adequate pain control.

Other comments:

The author states in the introduction there are 3 million annual deaths in Nigeria, yet numbers of patients reviewed at each site are low (4 -20 patients per week). Does the author have any insights into why patient numbers are low despite the estimated need being so high?

Line 94 - PC estimated to rise to what? the figure is missing

**Do you want your identity to be public for this peer review?** For information about this choice, including consent withdrawal, please see our Privacy Policy

Reviewer #1: No

Reviewer #2: **Yes: ** Kathryn Richardson

---

## [Decision Letter · Decision Letter 1]

1 May 2025

Organizational models and patient-reported outcomes for palliative care across five tertiary hospitals in Nigeria: an environmental scan

PGPH-D-24-02428R1

Dear Dr. Ogbenna,

We are pleased to inform you that your manuscript 'Organizational models and patient-reported outcomes for palliative care across five tertiary hospitals in Nigeria: an environmental scan' has been provisionally accepted for publication in PLOS Global Public Health.

Best regards,

Sakib Burza, MBChB, MRCGP, MSc, PhD

Academic Editor

Please include the minor comments from Reviewer 1, which I agree with.

Reviewer Comments (if any, and for reference):

Reviewer's Responses to Questions

**Comments to the Author**

Reviewer #1: (No Response)

publication criteria?

Reviewer #1: Yes

3. Has the statistical analysis been performed appropriately and rigorously?

Reviewer #1: Yes

4. Have the authors made all data underlying the findings in their manuscript fully available (please refer to the Data Availability Statement at the start of the manuscript PDF file)?

Reviewer #1: Yes

5. Is the manuscript presented in an intelligible fashion and written in standard English?

Reviewer #1: Yes

Reviewer #1: I would like to thank the authors for their careful and thoughtful responses to the feedback I shared. This is an important piece of work, and I am eager to see it published.

In my assessment the manuscript is ready for publication after the following small revision:

Line 193-197: Thank you for the revisions regarding the qualitative analysis; they are helpful. I understand this was not the main component of your study, but I think there is need for some further clarification.

The way you use the term “themes” in this section is not typical, unless you are referring to a specific methodology that I am not familiar with. When you say that the “themes were identified for each response”, could you define and describe the themes? It seems from the table on page 14 that you used the CFIR framework to code the data perhaps? If so, that is a reasonable deductive approach for this kind of study it would just help to clarify.

**Do you want your identity to be public for this peer review?** For information about this choice, including consent withdrawal, please see our Privacy Policy

Reviewer #1: No
